# Big Data and Predictive Analytics for Business Intelligence: A Bibliographic Study (2000–2021)

**Yili Chen [1], Congdong Li [1,2,\*] and Han Wang [3,4,5,\*]** 

1   School of Business, Macau University of Science and Technology, Avenida Wai Long, Taipa, Macao 999078, China
2   The School of Management, Jinan University, Zhuhai 519000, China
3   The Faculty of Data Science, City University of Macau, Macao 999078, China
4   The Department of Artificial Intelligence and Big Data Applications, Zhuhai Institute of Advanced Technology Chinese Academy of Sciences, Zhuhai 519000, China
5   The School of Computer, College of Beijing University of Technology Zhuhai, Zhuhai 519000, China
\*   Correspondence: licd@jnu.edu.cn (C.L.); wanghan@ziat.ac.cn (H.W.)

**Abstract:** Big data technology and predictive analytics exhibit advanced potential for business intelligence (BI), especially for decision-making. This study aimed to explore current research studies, historic developing trends, and the future direction. A bibliographic study based on CiteSpace is implemented in this paper, 681 non-duplicate publications are retrieved from databases of Web of Science Core Collection (WoSCC) and Scopus from 2000 to 2021. The countries, institutions, cited authors, cited journals, and cited references with the most academic contributions were identified. Social networks and collaborations between countries, institutions, and scholars are explored. The cross degree of disciplinaries is measured. The hotspot distribution and burst keyword historic trend are explored, where research methods, BI-based applications, and challenges are separately discussed. Reasons for hotspots bursting in 2021 are explored. Finally, the research direction is predicted, and the advice is delivered to future researchers. Findings show that big data and AI-based methods for BI are one of the most popular research topics in the next few years, especially when it applies to topics of COVID-19, healthcare, hospitality, and 5G. Thus, this study contributes reference value for future research, especially for direct selection and method application.

**Keywords:** big data; predictive analytics; business intelligence; bibliographic study; CiteSpace

## 1. Introduction

Business intelligence (BI) is related to insights extracted from the information of companies and marketing, based on which, a strategic decision-making process is implemented for firm development and marketing enhancements [1]. It is estimated that BI-related industries occupy the largest share of global business investments pertaining to information technology (IT) [2–4]. Predictive analysis is one of the most significant processes of BI performing. It aims to support managers to make reasonable decisions by predicting future development trends based on historical data [5]. Analytic tools and technologies are widely developed for expectation forecasting and business strategy simulation, where statistical modeling, mathematical calculation, result simulation, and finding visualization are included [4,5].

With more and more data from the company's internal and external platforms, big data analysis has become the method with the most potential for BI insight extraction [1,6,7]. In particular, with the advanced technology of artificial intelligence (AI) showing significant advantages of high efficiency, high accuracy, time-saving and resource saving, the intensity of BI prediction analysis has been significantly enhanced [6,8–10]. Open-source analytics tools based on deep learning and machine learning are accessible and widely applied to the

business decision-making processes [11], such as the tools of Microsoft Power BI [11,12], Google Analytics [13], SETLBI [14] and the accessible models of GitHub Repositories [15].

Scientometrics is a quantitative analytic subject for academic literature analysis [16–19]. It plays a vital role in research evaluation and disciplinary insight exploration. It is responsible for the identification of core journals, authors, institutions, countries, citations, topics, and historical trends, as well as future development forecasting [16,20]. CiteSpace is one of the most popular tools for scientometrics study. It presents a great performance on historic developing trend exploration, hotspot mapping, social network visualization, and analysis index calculation [18,20,21].

This study delivers a bibliographic study by utilizing a citation exploration tool of CiteSpace on 681 non-duplicate citations from WoSCC and Scopus, where 364 and 350 original publications are involved from WoSCC and Scopus respectively. The most academic influential countries, institutions, authors, and citations are recognized. Core journals, hotspot matric, developing trends, and disciplinaries are discussed. The structure of this study is organized as the following. Section 2 is related to the methodology this article employed, results and discussion is presented in Section 3. A conclusion is involved in Section 4.

## 2. Methodology

### 2.1. Data Source

This study collects literature from datasets of Web of Science Core Collection (WoSCC) and Scopus. These two databases are involved as the major citation sources for bibliographic studies [22–24]. The search string in WoSCC is set as: TS = ("business intelligen *" OR "BI") AND ("predict *" OR "forecast" OR "foresee") AND ("big data"). The search string in Scopus is set as: TS = ("business intelligence" OR "business intelligent" OR "BI") AND ("predict" OR "prediction" OR "forecast" OR "foresee") AND ("big data"). Document type = ("article" or "review"). Time span = (from "1 January 2000" to "7 November 2021").

### 2.2. Analysis Tools

This study utilizes the tool of CiteSpace (5.3.R4, 64-bit) [25,26] and JRE (1.8) [27] for the literature analysis, accessed on 4 May 2020. This software could be downloaded from the website of https://sourceforge.net/projects/citespace/ (accessed on 18 August 2022), it is generated by Chen etc. [25] According to the "The CiteSpace Manual" released in 2014 [28], CiteSpace I [29] and CiteSpace II [25] are the first and the second version of this tool. The initial publication of CiteSpace I [29] and CiteSpace II [25] from Chen has been cited on Google Scholar 1882 and 4347 times, respectively (The retrieval time is 16 September 2022). CiteSpace is interactive software running based on JRE (1.8) environment, aiming to knowledge extraction, exploring academic achievements, in-depth knowledge graphic visualization, scientific review, and literature quantitative analysis [25,26,28,30]. Developing trend of academic opinions based on time series, contributed scholars, institutions, journals, countries, and discipline subjects are able to be identified and analyzed by using this software, which has widely been utilized in bibliographic studies [18,21,23,24,26]. CiteSpace generates social networks with nodes and links, which indicates the degree of cooperation between authors, institutions, and countries [20]. The shape of the nodes reflects the influential degree of the author, citation, journal, institution, country, etc. The weight of lines represents the degree of betweenness among nodes. The centrality value reflects the significant degree of nodes, where nodes with a centrality $\geq 0.1$ are regarded as the key nodes [20,31].

In this study, the keyword mapping, historic trend, and cluster classification are displayed. The major authors (according to the centrality), core journals (according to the centrality of publications), major institutions (according to the number of publications), most influential counties (according to the centrality of publications), most contributed papers (according to the number of publications), key topics (according to the number

of publications), and major involved category (according to the number of publications) are identified.

## 3. Results and Discussions

### 3.1. Trends in the Literature

The historic developing trend of publications related to the topic of big data and predictive analytics applied to BI is presented in Figure 1. As Figure 1 shows, the total number of non-duplicate literature from WoSCC and Scopus is in an increasing trend by 2011, this trend seems to be continuing in the next few years. The same trend occurred in WoSCC and Scopus databases, where the developing trend is more significant for Scopus after 2015. Besides, before 2019, the number of papers from WoSCC is more than Scopus, the situation is reversed in 2020.

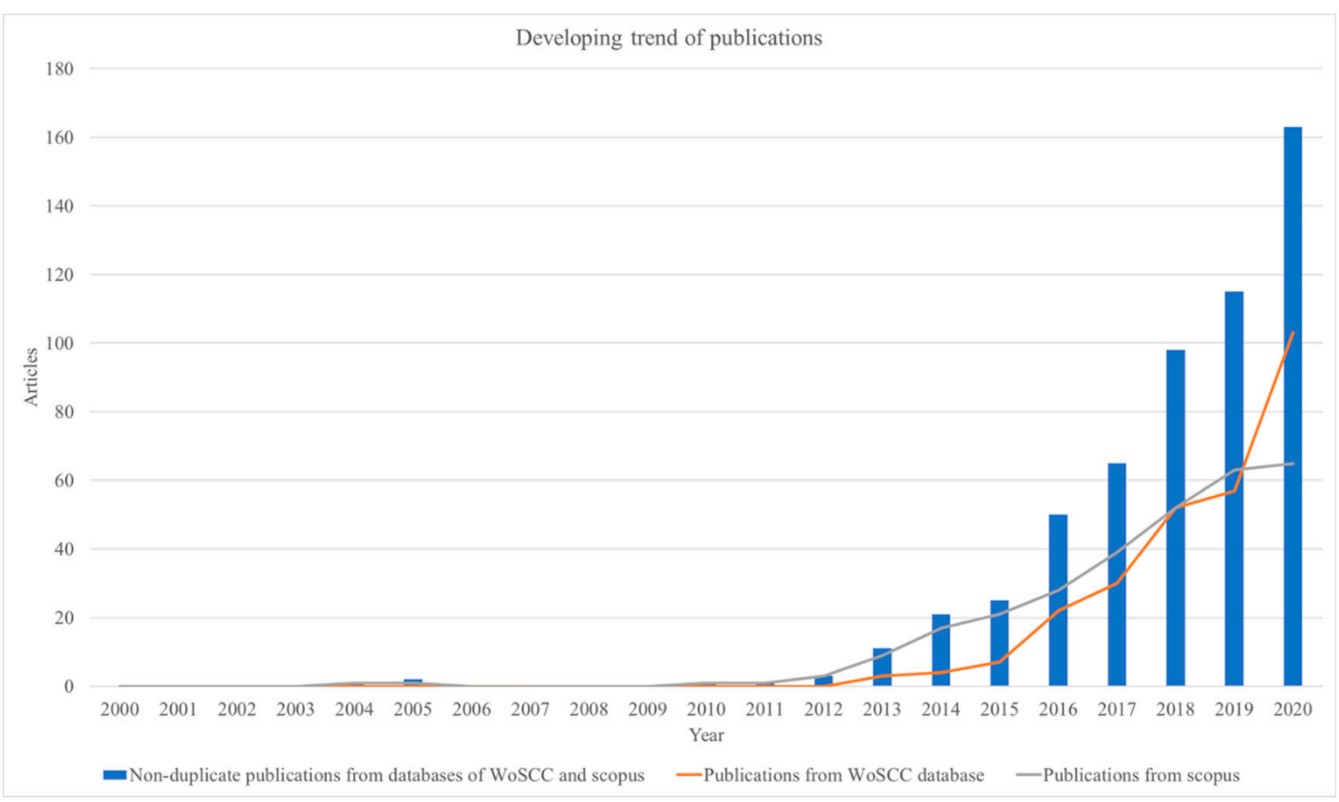

**Figure 1.** History trend analysis of publications.

### 3.2. Analysis of Countries and Institutions

A country distribution map is generated in Figure 2, with the top 5 academic contributed countries ranked by the number of publications, as listed the Table 1. Based on the figure and table mentioned, findings show that the USA presents a significant advantage in the research of big data and forecasting analytics for BI, with the count and centrality value of publications of 93 and 0.1. Countries of India (count = 76, centrality = 0.08), China (count = 71, centrality = 0.02), England (count = 54, centrality = 0.11) and Germany (count = 38, centrality = 0.04) are listed as the second, third, fourth and fifth influential countries. Moreover, with 63 notes and 241 links (density = 0.1234) displayed in Figure 2, the cooperation between countries is relatively weak and should be paid attention to.

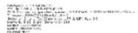

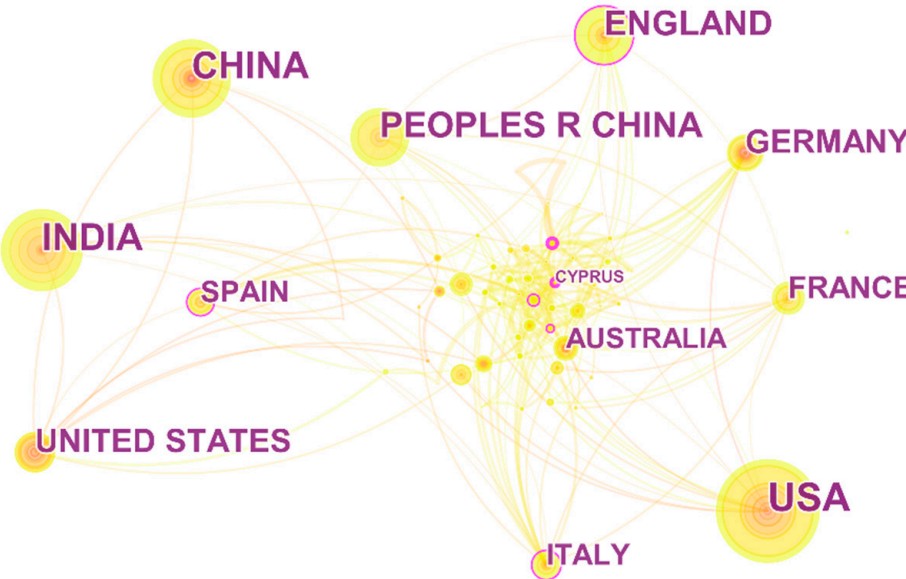

**Figure 2.** The map distribution of countries (The yellow points represent contributed countries).

**Table 1.** Top 5 countries according to the number of publications.

| Ranking | Country | Count | Centrality |
|---|---|---|---|
| 1 | USA/The United States | 93/39 | 0.1/0.09 |
| 2 | India | 76 | 0.08 |
| 3 | China/Republic of China | 71/55 | 0.02/0.03 |
| 4 | England | 54 | 0.11 |
| 5 | Germany | 38 | 0.04 |

As Figure 3 and Table 2 shown, the distribution of institutions is exhibited, sorted by the number of publications. The most contributed institution is the department of Computer Science, University of Nevada, Las Vegas, NV, the United States (count = 16), followed by Swansea University, UK (count = 6), the University College London (UCL) (count = 6), Nanjing University, China (count = 5), and National Institute of Industrial Engineering (NITIE), Maharastra (count = 4). Otherwise, with 119 notes and 190 links (density = 0.0271) illustrated in Figure 3, the collaboration between institutions is weak, and should be focused on in the future.

**Table 2.** Top 5 institutions according to the number of publications.

| Ranking | Institutions | Count | Centrality |
|---|---|---|---|
| 1 | Department of Computer Science, University of Nevada, Las Vegas, NV, the United States. | 16 | 0 |
| 2 | Swansea University, UK | 6 | 0 |
| 3 | University College London (UCL) | 6 | 0 |
| 4 | Nanjing University, China | 5 | 0 |
| 5 | National Institute of Industrial Engineering (NITIE), Mumbai, Maharastra | 4 | 0 |

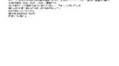

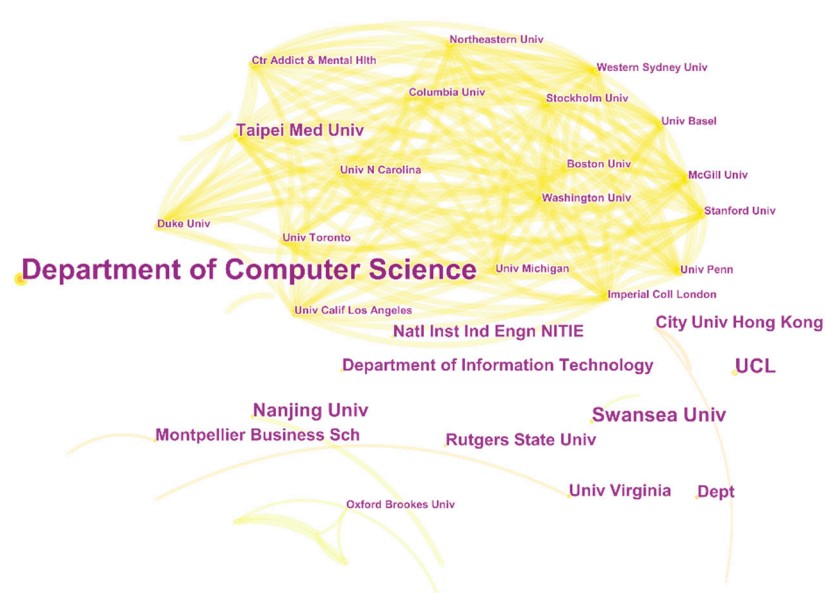

**Figure 3.** The distribution of institutions (The yellow links represent collaboration relationship between institutions).

### 3.3. Analysis of Cited Journals, Cited Authors, and Cited References

The distribution of cited journals is presented in Figure 4, with the top 5 cited journals listed in Table 3, according to the centrality value. The impact factor (IF) is obtained from an IF search engine of Resurchify (https://www.resurchify.com/ (accessed on 18 August 2022)), the access date is 7 November 2021. Findings show that almost all the journals are the top list journal with a relatively high IF. The core journal is identified as Management Science (count = 82, centrality = 0.12, IF = 5.04), followed by MIS Quarterly (count = 130, centrality = 0.09, IF = 7.198), Harvard Business Review (count = 103, centrality = 0.09, IF = 1.66), Decision Support Systems (count = 124, centrality = 0.08, IF = 7.04), and European Journal of Operational Research (count = 83, centrality = 0.07, IF = 6.02).

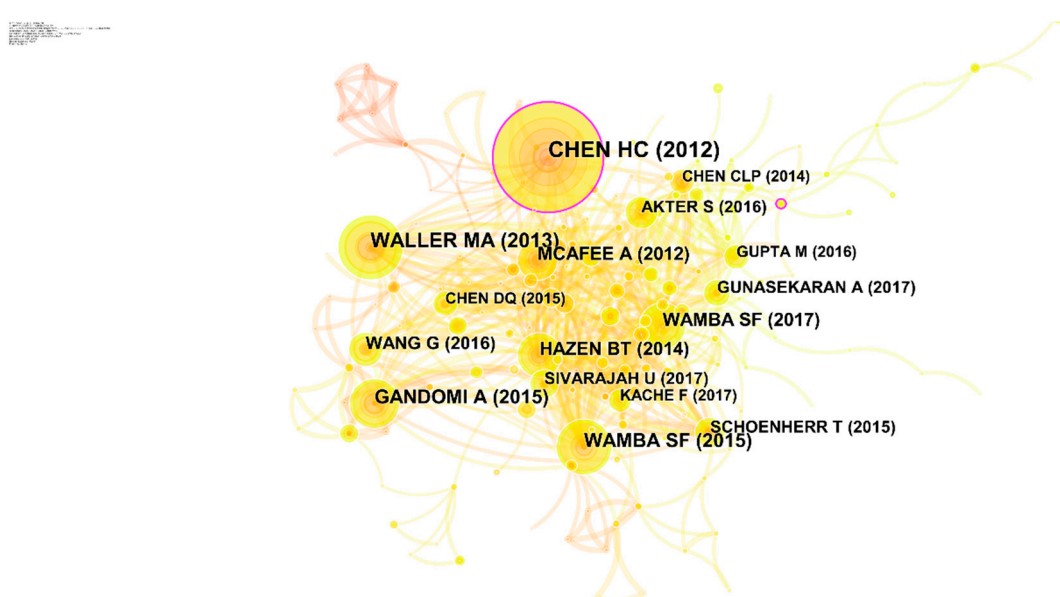

**Figure 4.** The distribution of cited journals (The yellow links represent citation relationship between journals).

**Table 3.** The top 5 co-cited journals according to centrality.

| Ranking | Journals | Count | Centrality | Impact Factor (2021) |
|---------|----------|-------|------------|----------------------|
| 1 | Management Science | 82 | 0.12 | 5.04 |
| 2 | MIS Quarterly | 130 | 0.09 | 7.198 |
| 3 | Harvard Business Review | 103 | 0.09 | 1.66 |
| 4 | Decision Support Systems | 124 | 0.08 | 7.04 |
| 5 | European Journal of Operational Research | 83 | 0.07 | 6.02 |

The distribution of cited authors is displayed in Figure 5. According to the centrality value, the top 5 authors and their representative works are listed in Table 4. The most influential scholar is regarded as Davenport, T.H. (count = 71, centrality = 0.1), his work "Big data: the management revolution" [32] is released in 2012. This article proposes several valuable BI strategies for the company decision-making process, which is widely cited by other researchers (cited by 5735 times according to Google Scholar, the search date is 7 November 2021). The works of other four top scholars are all related to the topic of big data analysis and BI [32–35]. Besides, with 341 nodes and 974 links involved in Figure 5, it is indicated that cooperation between scholars should be encouraged.

**Table 4.** Top 5 authors according to the centrality.

| Ranking | Authors | Count | Centrality | Involved Publications |
|---------|---------|-------|------------|-----------------------|
| 1 | Davenport, T.H. | 71 | 0.1 | Big data: the management revolution [32]. |
| 2 | Chiang, Roger HL | 5 | 0.07 | Strategic value of big data and business analytics [33]. |
| 3 | Dubey, Rameshwar | 45 | 0.06 | Education and training for successful careers in big data and business analytics [34]. |
| 4 | McAfee, Andrew | 39 | 0.06 | Big data: the management revolution [32]. |
| 5 | Davenport, Thomas H | 20 | 0.06 | Data scientist [35]. |

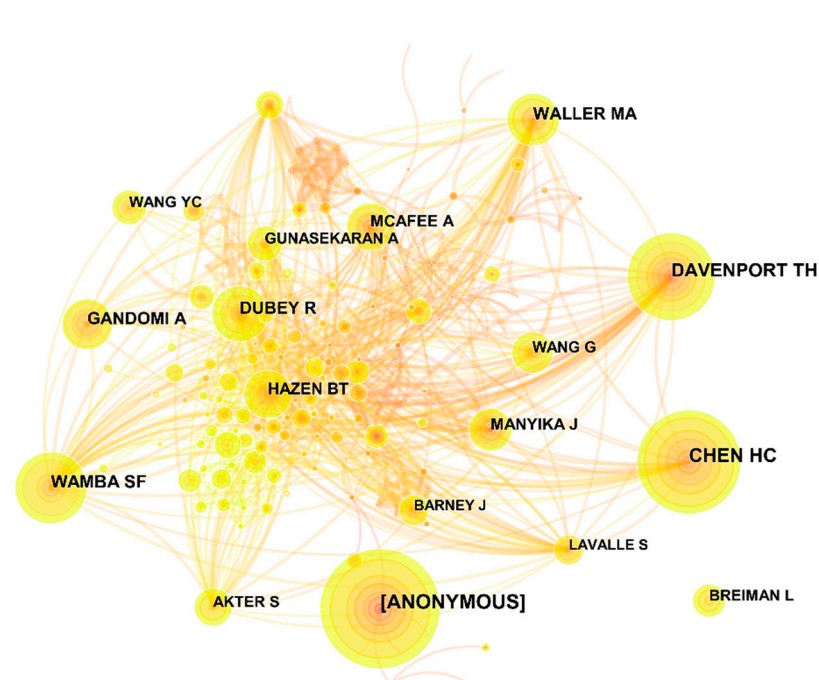

**Figure 5.** The distribution of cited authors (The links represent collaboration relationship between authors).

The distribution of cited references is illustrated in Figure 6. The Top 5 references are ranked in Table 5. The most influential article is "Business intelligence and analytics: From big data to big impact" (count = 71, centrality = 012), which is cited by 6376 articles according to google scholar, the search date is 7 November 2021.

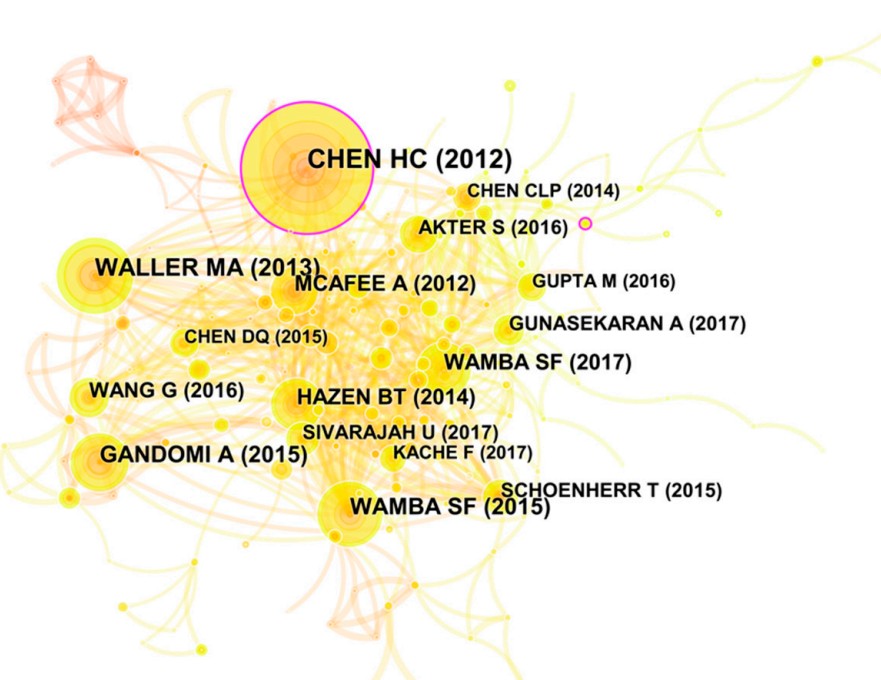

**Figure 6.** The distribution of cited references (The links represent citation relationship between references).

**Table 5.** Top 5 cited references according to the number of being cited.

| Ranking | Count | Centrality | Year | Cited Reference | Number of Cited by (According to Google Scholar) |
|---|---|---|---|---|---|
| 1 | 71 | 0.12 | 2012 | Business intelligence and analytics: From big data to big impact [36] | 6376 |
| 2 | 45 | 0.05 | 2013 | Data science, predictive analytics, and big data: a revolution that will transform supply chain design and management [37]. | 1336 |
| 3 | 38 | 0.09 | 2015 | How 'big data' can make big impact: Findings from a systematic review and a longitudinal case study [38]. | 1407 |
| 4 | 34 | 0.07 | 2015 | Beyond the hype: Big data concepts, methods, and analytics [38,39] | 3929 |
| 5 | 31 | 0.05 | 2017 | Big data analytics and firm performance: Effects of dynamic capabilities [40]. | 986 |

*3.4. Analysis of Categories, Hotspots, and Burst Topic Historic Trends*

The distribution of disciplinaries is exhibited in Figure 7. The top 10 categories are listed in Table 6. It is clustered into the following categories, which are big data, business, management and economics, computer science, interdisciplinary applications, machine learning, economics, engineering, computer science, big data analytics, artificial intelligence, and mathematics. Seven of the ten disciplines are data science, computer science, mathematics, and engineering-based categories. Two of the ten disciplines are involved

in economy, business, and management. A category of interdisciplinary applications is uniquely included. Furthermore, with 50 nodes and 165 links (density = 0.1347) exhibited in Figure 7, it indicated that the interdisciplinary cross-field subjects should be paid more attention to, especially for the application of computer science to business and economics.

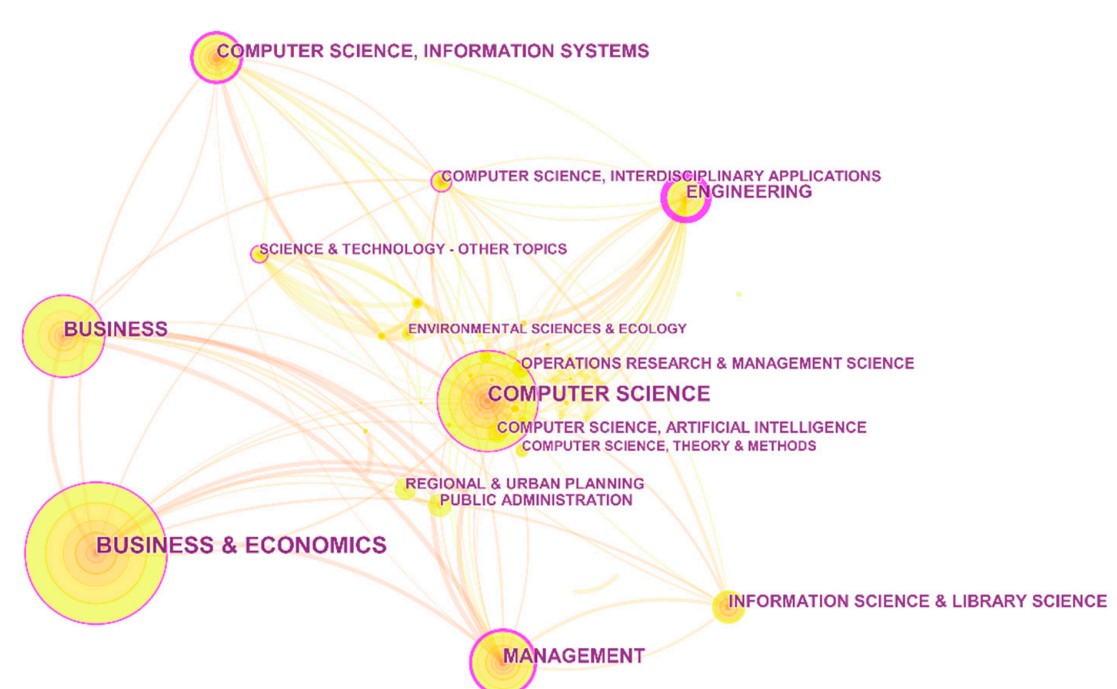

**Figure 7.** The distribution of categories (The circles represent categories involved).

**Table 6.** Top 10 categories distribution.

| Ranking | Category | Count | Centrality |
|---|---|---|---|
| 1 | Big data | 111 | 0.26 |
| 2 | Business, management & Economics | 207 | 0.25 |
| 3 | Computer science, Interdisciplinary applications | 23 | 0.13 |
| 4 | Machine learning | 24 | 0.12 |
| 5 | Economics | 13 | 0.12 |
| 6 | Engineering | 42 | 0.11 |
| 7 | Computer science | 102 | 0.1 |
| 8 | Big data analytics | 44 | 0.1 |
| 9 | Artificial intelligence | 74 | 0.09 |
| 10 | Mathematics | 46 | 0.04 |

The distribution of keywords is displayed in Figure 8. The top 10 keywords are listed in Table 7, ranked by the number of publications. Keywords of big data, artificial intelligence, machine learning, business intelligence, data mining, predictive analytics, forecasting, big data analytics, decision making, and data analytics are included.

**Table 7.** Top 10 keywords ranking according to the number of publications involved.

| Ranking | Keywords | Count | Centrality |
|---|---|---|---|
| 1 | big data | 368 | 0.14 |
| 2 | artificial intelligence | 160 | 0.25 |
| 3 | machine learning | 109 | 0.03 |
| 4 | business intelligence | 99 | 0.07 |
| 5 | data mining | 80 | 0.23 |
| 6 | predictive analytics | 78 | 0.26 |
| 7 | forecasting | 75 | 0.03 |
| 8 | big data analytics | 70 | 0.13 |
| 9 | decision making | 67 | 0.14 |
| 10 | data analytics | 64 | 0.08 |

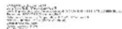

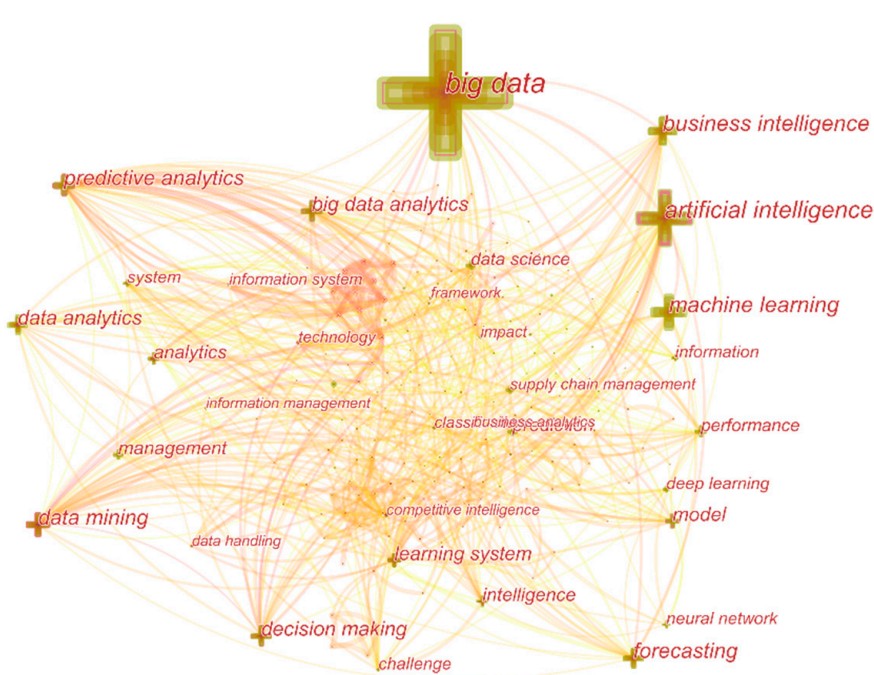

**Figure 8.** The distribution of keywords (The cross pattern represents the keyword).

The time zone of the hotspot is exhibited in Figure 9, and the developing history of keywords and cluster visualization is illustrated in Figure 10. Findings show that data mining is the first occurred keyword (2012). It is followed by technology, decision making, big data, large amounts of data, management science, business process, software reliability, and decision support system in 2013, where the topic of decision making, and big data are the strongest burst topic during the entire developing history. Keywords of business intelligence, social networking (online), forecasting, AI, digital storage, competitive intelligence, data analytics, competition, information system, predictive analytics, and weather forecasting burst in 2014. Keywords of analytics model, big data analytics, commerce, learning system, machine learning, data handling, administrative data processing, and manufacturing appeared in 2015. Keywords of information performance intelligence, algorithm, sale prediction, information analysis, distributed computer system, metadata, predictive modeling, data visualization, support vector machine (SVM), privacy, regression, and demand regression analysis emerged in 2016. Keywords of system, data science, management, learning algorithm, classification, efficiency, information, management, health care, machine learning technique, predictive model, quality management, big data application, personality knowledge, social media, dynamic capability, knowledge management, competitive advantage, perspective selection and risk burst in 2017. In 2018,

keywords of design, internet, neural network (NN), random forest (RF), machine learning (ML) model, deep learning (DL), smart city, knowledge discovery, risk assessment, internet of things (IoT), and social network occurred. In 2019, AI, intelligent computing, time series, advanced analytics, literature review, crime, data technology, decision-making, big data analytics, customer satisfaction, and innovation burst. Hotspots of service, user acceptance, industry 4.0, and sentiment analysis appeared in 2020. In the latest year of 2021, the keywords of AI, COVID-19, information technology (IT), healthcare, data analysis, 5G, mobile communication system, hospitality, text mining, and satisfaction burst in publications.

Furthermore, the keywords are clustered into 8 classifications, which are building mature BI (model), meaningful insights, mining shopper data stream, predictive model, event-based prediction, using big data, traveler-generated content, probabilistic electric load forecasting, and research needs.

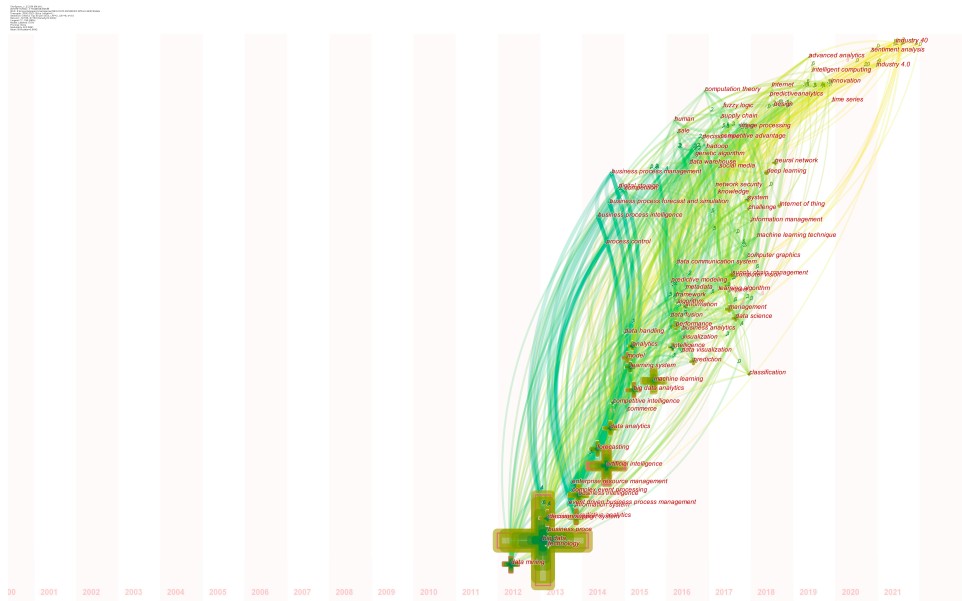

**Figure 9.** The time zone of hotspots (The cross pattern represents the keyword).

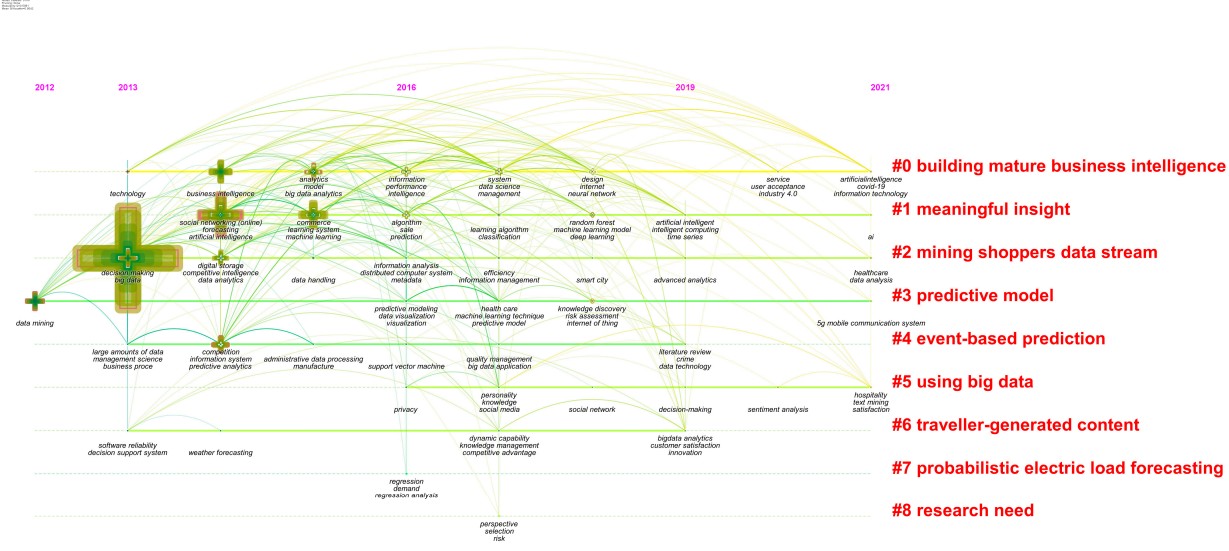

**Figure 10.** Developing a history of keywords and cluster visualization (The cross pattern represents the keyword).

The 17 keywords with the strongest citation bursts are listed in Figure 11. Based on the timeline, burst keywords are listed as follows: data mining (2012–2013), business process (2013–2014), social networking (online) (2014–2015), enterprise resource management (2014–2015), business process intelligence (2014–2015), learning system (2015–2018), data handling (2015–2019), information analysis (2016–2017), intelligent system (2016–2017), privacy (2016–2018), data visualization (2016–2017), sale (2016–2017), business analytics (2016–2019), personality (2017–2018), decision tree (DT) (2018–2019), competitive advantage (2018–2019), intelligent computing (2019–2021).

Thus, results show an increasing trend of publications on the topic of big data and predictive analytics applicated to BI. The study identifies the most contributed countries and institutions. Core journals, scholars, and references are recognized. Categories, hotspots, and citation burst history are explored. Besides, the social network and cooperation relationship between countries, institutions, and authors, as well as the cross degree of involved disciplinaries, are analyzed in this section.

View Citation Burst History

# Top 17 Keywords with the Strongest Citation Bursts

| Keywords | Year | Strength | Begin | End | 2000 - 2021 |
|---|---|---|---|---|---|
| data mining | 2000 | 3.2296 | 2012 | 2013 | |
| business proce | 2000 | 3.0941 | 2013 | 2014 | |
| social networking (online) | 2000 | 2.2396 | 2014 | 2015 | |
| enterprise resource management | 2000 | 3.135 | 2014 | 2015 | |
| business process intelligence | 2000 | 3.135 | 2014 | 2015 | |
| learning system | 2000 | 3.8534 | 2015 | 2018 | |
| data handling | 2000 | 2.3587 | 2015 | 2019 | |
| information analysis | 2000 | 3.4009 | 2016 | 2017 | |
| intelligent system | 2000 | 2.9136 | 2016 | 2017 | |
| privacy | 2000 | 2.5549 | 2016 | 2018 | |
| data visualization | 2000 | 2.9136 | 2016 | 2017 | |
| sale | 2000 | 2.4275 | 2016 | 2017 | |
| business analytics | 2000 | 2.8593 | 2016 | 2019 | |
| personality | 2000 | 2.3956 | 2017 | 2018 | |
| decision tree | 2000 | 2.93 | 2018 | 2019 | |
| competitive advantage | 2000 | 2.5683 | 2018 | 2019 | |
| intelligent computing | 2000 | 2.8562 | 2019 | 2021 | |

**Figure 11.** The top 17 citation burst history of keywords (The red part represents the burst period).

*3.5. Discussions*

The applications of big data and predictive analytics to BI is a significant research topic during the last 10 years. It is estimated that the interest in this field will be continually raising in the next few years. The U.S. and the institution of the University of Nevada (USA) are recognized as the most academically influential country and institution, respectively. However, cooperation between countries and institutions should be encouraged.

The journal of Management Science, author of Davenport, T.H., and the article "Business intelligence and analytics: From big data to big impact" [36] are identified as the most academically significant in this research field. Based on the result of category identification, this study concluded that most of the articles are from the disciplinary of data science, computer science, and engineering, less from the disciplinaries of business, economy, and management subjects. The reason could be technical barriers to big data, AI, and predictive analytics are relatively higher for scholars from economic, business, and management

subjects. Thus, the cooperation between scholars, especially authors from different research areas, should be encouraged to cooperate for further research.

Furthermore, according to the results of keyword mapping and topic bursting history trend, the burst history of big data and predictive analytic related methodologies, BI-based applications, and the major challenges, as well as the latest hot topics, are identified and listed as the following.

Firstly, the developing trend of big data and AI-based methods could be concluded as the following, i.e., data mining (burst in 2012), digital storage (burst in 2013), predictive analytics (burst in 2014), ML (burst in 2015), distributed computer system (burst in 2016), predictive modeling and visualization (burst in 2016), SVM (burst in 2016), regression (burst in 2016), classification algorithms (burst in 2017), NN (burst in 2018), RF (burst in 2018), DT (burst in 2018), DL (burst in 2018), prediction algorithms based time-series (burst in 2019), sentiment analysis (burst in 2020), and text mining (burst in 2021). The representative citations of each technology in the specific burst point of time are explored. Contributions of references are illustrated in Table 8.

Secondly, the specific applications of these methods to BI could be summarized as the following, which are decision making (burst in 2013), business process intelligence (burst in 2013), competitive intelligence and competition analysis (burst in 2014), enterprise resource management (burst in 2014), administrative management (burst in 2015), commerce enhancement (burst in 2015), manufacture development (burst in 2015), sale prediction (burst in 2016), information management (burst in 2017), quality management (burst in 2017), knowledge management (burst in 2017), risk assessment (burst in 2018), customer satisfaction management (burst in 2018), service improvement (burst in 2020), user acceptance development (burst in 2020) and satisfaction improvement (burst in 2021). The developing trend of BI applications based on big data is presented in Table 9, findings of representative references are discussed in the table.

**Table 8.** Developing trend exploration of methods.

| Burst Year | Methods | References | Conclusions |
|---|---|---|---|
| 2012 | data mining | Open business intelligence: on the importance of data quality awareness in user-friendly data mining [41]. | A highly qualified guiding mechanism of data mining of linked open data is necessary for open business intelligence, especially for non-expert users. |
| 2013 | digital storage | Business Process Analytics Using a Big DataApproach [42]. | Based on Hbase and Apache Hadoop, big data analytics in distributed environments present great advantages for business performance management. |
| 2014 | predictive analytics | Big data and predictive analytics in ERP systems for automating decision making processes [43]. | By identifying potential risks and opportunities, big data predictive analytics in enterprise resource planning (ERP) plays a great role in automating the decision-making process. |
| 2015 | ML | Efficient Machine Learning for Big Data: A Review | ML is responsible for decision making processes for BI, where efficient sustainable data modeling is necessary for big data processing. |
| 2016 | distributed computer system | Business-intelligence mining of large decentralized multimedia datasets with a distributed multi-agent system [44]. | Agent-oriented modeling techniques are novel solutions to meet the distributed data-mining processes. |
| 2016 | SVM | Big data analytics in healthcare: A survey approach [45]. | SVM is a typical ML algorithm for big data analytics in the BI area, which is responsible for classifying data into binomial classes or multilevel classes. |

**Table 8.** *Cont.*

| Burst Year | Methods | References | Conclusions |
|---|---|---|---|
| 2016 | regression | A comparative analysis on linear regression and support vector regression [46]. | Regression algorithms play a great role in BI research, which is mostly utilized for time-series data analytic tasks. |
| 2017 | classification algorithms | Knowledge management for business intelligence measurement in an e-business system [47]. | Algorithms of hierarchical ascendant classification and product classification present a great strength for BI knowledge management. |
| 2018 | NN | Deep learning architecture for high-level feature generation using stacked auto encoder for business intelligence [48]. | NN-based deep learning model delivers great performance for big data-based BI tasks. |
| 2018 | RF | | |
| 2018 | DL | Deep learning architecture for high-level feature generation using stacked auto encoder for business intelligence [48]. | Compared to ML algorithms, DL strategies show a great advantage for high-level representation extractions. |
| 2019 | prediction algorithm-based time-series | Large Multivariate Time Series Forecasting: Survey on Methods and Scalability [49]. | Forecasting models play a vital role in BI time series data analysis, where the prediction model optimization of selection, dimension reduction, and shrinkage are highlighted. |
| 2020 | sentiment analysis | Big data and sentiment analysis: A comprehensive and systematic literature review [50]. | Sentiment analysis based on textual big data is helpful for BI enhancements in aspects of efficiency, flexibility, and intelligence. |
| 2021 | text mining | Research trends on big data domain using text mining algorithms [51]. | Text mining based on big data for clustering and association evaluations is helpful for BI management, modern techniques like cloud computing, green information, and open source should be considered in future research. |

**Table 9.** Developing trends of BI applications based on big data.

| Burst Year | BI Applications | References | Conclusions |
|---|---|---|---|
| 2013 | decision making | Data science and its relationship to big data and data-driven decision-making [52]. | Data-driven decision-making process has the potential for maintaining BI sustainability, especially for large-scale datasets. However, an explicit research design based on the fundamental of business management is necessary. |
| 2013 | business process intelligence | Business process analytics using a big data approach [42]. | Big data analytics in a distributed environment is a key solution for evidence-based business process management (BPM), especially to meet high requirements of low cost, high quality, and timely measurement. |
| 2014 | competitive intelligence and competition analysis | Research on Enterprise Competitive Intelligence Development and Strategies in the Big Data Era [53]. | Big data-driven strategies based on deep insight exploration are key for enterprise competitive intelligence development, especially for company organization, resource sharing, collaborations, and security protection. |
| 2015 | administrative management | Impact of ICT on administrative management processes [54]. | With the development of ICT, electronic administrative management is enhanced, and participants are able to be involved in the decision-making process directly. |

**Table 9.** *Cont.*

| Burst Year | BI Applications | References | Conclusions |
| --- | --- | --- | --- |
| 2015 | commerce enhancement | Big data-based system model of electronic commerce [55]. | By analyzing performances of customer behaviors, deliveries, sales, marketing, competitors, and payments, the big data-based e-commerce model shows a great advantage for online store management. |
| 2015 | manufacture development | Application of business intelligence solutions on manufacturing data [56]. | Analysis based on manufacturing data is an efficient way to generate strategic reports and enhance manufactory efficiency. |
| 2016 | sale prediction | Prediction of sales using Big data analytics [57]. | With big data solutions of Apache Flume, hive. And HDFS, analysis of smart data is an effective way for purchase intention exploration, which is significant for marketing enhancements. |
| 2017 | information management | Improving Governance of Integrated Reservoir and Information Management Leveraging Business Process Management and Workflow Automation [58]. | Information management plays an important role in business process management, which is responsible for process performance metric tracking, communicating value enlargement, strategic decision-making, etc. |
| 2017 | quality management | Deep-level quality management based on big data analytics with case study [59]. | Deep-level quality management based on process large-scale data |
| 2017 | knowledge management | Towards integrated models of big data (BD), business intelligence (BI) and knowledge management (KM) [60]. | An integrated pattern of big data, business intelligence, and acknowledgment management is an advanced solution for competitive advantage enhancements. |
| 2018 | risk assessment | Adaptive management approach for more availability of big data business analytics [61]. | Through forecasting and measuring analysis patterns, adaptive risk assessment methods based on a big data environment present a great advantage to meet the requirements of time-consuming and high accuracy. |
| 2018 | customer satis- faction management | Advanced customer analytics: Strategic value through integration of relationship-oriented big data [62]. | Big data-based customer analytics deliver the potential for companies' sustainable competitive advantages. |
| 2020 | service improvement | The application of a business intelligence tool for service delivery improvement: The case of South Africa [63]. | BI-based decision supporting system (BIDSS) model is responsible for service improvement, big data analytics on users' feedback and service delivery is a key method for BIDSS building. Quick response, quality service delivery, transparency, and accessibility are the key aspects of service improvement. |
| 2020 | user acceptance development | Understanding user acceptance of blockchain-based smart locker [64]. | The key factors that influence user acceptance of new technology-oriented products are function, convenience, and security insurance. |
| 2021 | satisfaction improvement | Quality Big Data Analysis and Management Based on Product Satisfaction Index [65]. | Big data analysis and product satisfaction management are effective solutions for customer relationships and quality performance management. |

Thirdly, it is indicated that the issues of software reliability (burst in 2013), privacy (burst in 2016), and personal information (burst in 2017) are the most focused challenges of

this research field. The burst trend of challenges is discussed in Table 10, and findings and conclusions are explored in this table.

**Table 10.** Challenge burst trend explorations.

| Burst Year | Challenges | References | Conclusions |
|---|---|---|---|
| 2013 | software reliability | Cloud solution in Business Intelligence for SMEs–vendor and customer perspectives [66]. | Reliability and cost are the core issues for BI analytics. |
| 2016 | privacy | CRSA cryptosystem based secure data mining model for business intelligence applications [67]. | Privacy and authenticity of datasets are significant issues for BI applications. Solutions of secure and privacy preserved mining models are responsible for resource and time saving and high accuracy, preventing. |
| 2017 | personal information | Risk magnification framework for clouds computing architects in business intelligence [68]. | Control system of sensitive and personal data is necessary for BI management, especially for a distributed cloud computing environment, which is significant for the maintenance of security, reliability, and compliance. |

Moreover, as the Table 11 shown, in the latest year of 2021, the hot topic of COVID-19, healthcare, hospitality, and 5G are the data source and practical applications for big data, predictive analytics, and BI research. COVID-19 inevitably influents almost all the business industries, especially in the latest 2 years. It brings rapid development growth to the healthcare industry. However, the situation is the opposite for hospitality. BI-related strategies are regarded as a booster for the healthcare industry developing, as well as a useful solution for industrial recovery. The need for high speed and bandwidth of the internet is raised, since more time we spend on the internet instead of social contact with people face to face. With the advantages of low latency and higher speed, capacity, and reliability, it is a novel solution for the enhancements of multiple business industries.

**Table 11.** The latest topics burst in 2021.

| Burst Year | Latest Topics | References | Conclusions |
|---|---|---|---|
| 2021 | COVID-19 | [69–71] | The major topics of COVID-19, big data, predictive analytics, and BI research falls on the challenges and BI solution for firms due to the epidemic. The other topic is the application of BI utilized in the research on the influential effects of COVID-19 on business industries. |
| 2021 | healthcare | [72–74] | The major topics are related to the challenges and advanced technologies of BI applied to healthcare industries. Patient data safety and its business application are incentive topics, which should be discussed in the future. |
| 2021 | hospitality | [75–77] | Hospitality is one of the most influential industries by COVID-19, where BI and information technology-driven solution is the most effective and novel methods for levering the hospitality growth trend. |
| 2021 | 5G | [78–80] | 5G technology brings new opportunities for BI in the aspects of quality service monitoring, effective decision-making, efficient operation management, etc. |

Therefore, with the development of AI-based (ML and DL) and big data-related (distributed computing, IoT, etc.) technologies, predictive analytics with the advantages of high effectiveness, high accuracy, and resource-saving plays an increasingly significant role in business intelligence. With the enhancement of social media, sentiment analysis and text

mining exhibit an advanced strength for BI, especially for customer relationship management. In 2021, COVID-19, the healthcare industry, hospitality, and 5G are recognized as the hottest topic in big data and predictive analytics to BI studies, it is indicated that this topic shows a great potential to be hotspots in the next years.

## 4. Conclusions

This study utilized the tool of CiteSpace to implement a bibliographic study on 681 non-duplicate citations retrieved from WoSCC and Scopus databases from 2000 to 2021, the research topic is related to the application of big data and predictive analytics to business intelligence. Findings show that the publications on this topic are at an increasingly developing trend, which is predicted to be continuing in the next few years. Besides, the most academic influential countries, institutions, journals, authors, and articles are identified in this study. Disciplinaries, hotspot metrics, and topic burst history trends are discussed. The social network between countries, institutions, authors, and categories is explored. The developing trend of methodologies, BI applications, and challenges related to big data, predictive analytics, and BI. The reason hot topics burst in 2021 is discussed. It contributes significant reference value for related researchers in the future, especially for the topic selection and method application.

Limitations are concluded as the following. Firstly, articles of WoSCC and Scopus only in English are involved in this study, other literature databases should be considered in future research. Secondly, the insights are extracted based on the results analyzed by the tool of CiteSpace, where papers are requested to be imported as the standard format, the information other than the imported citation is not able to be explored. Thirdly, without considering other measurement methods, this article identifies the contributed institutions, scholars, journals, and topics only at the academic level.

Thus, four pieces of advice are delivered for future research. Firstly, articles in more than one language from multiple databases are suggested to be analyzed. More keywords, like "data mining" [81,82], should be considered during the searching in databases. Secondly, different tools, including text mining tools, are encouraged for scientific article explorations [83,84]. Thirdly, when it comes to identifying the contributed institutions, scholars, journals, and topics, a practical perspective, such as economic and social contributions, should be explored, especially for the firms and managers. Fourthly, this study recommends a research direction for future research, which is that big data, predictive analytics, and BI could be considered applied to the industries related to COVID-19, healthcare, hospitality, and 5G. Explainable big data and AI approaches should be paid more attention, since, without a high level of interpretability, transparency, and accuracy, the black-box AI prediction algorithms may cause a huge economic loss. Finally, studies of the novel method of text mining based on social media data are suggested for BI enhancements.

**Author Contributions:** Conceptualization, Y.C. and H.W.; methodology and software, H.W.; investigation, Y.C. and C.L.; writing—original draft preparation, Y.C., C.L. and H.W.; writing—review and editing, Y.C. and H.W.; supervision, C.L.; funding acquisition, H.W. All authors have read and agreed to the published version of the manuscript.

**Funding:** This research was funded by the 2020 Key Technology R&D Program of Guangdong Province, grant number 2020B1111540001, Zhuhai Technology and Research Foundation, grant number ZH01110405180056PWC, Zhuhai Technology and Research Foundation, grant number ZH22036201210034PWC, and Zhuhai Basic and Application Research Project, grant number of ZH22017003200011PWC.

**Institutional Review Board Statement:** Not applicable.

**Informed Consent Statement:** Not applicable.

**Data Availability Statement:** Not applicable.

**Conflicts of Interest:** The authors declare no conflict of interest.

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
