# Peer review of "Big Data and Predictive Analytics for Business Intelligence: A Bibliographic Study (2000–2021)"

_forecasting, doi:10.3390/forecast4040042_

Round 1
Reviewer 1 Report
Overall interesting. Similar to some other studies. Minor comments mainly style related,
1. Introduction could flow better, jumps.
2. Omit last para sect. 2.1
3. Fig 1 Omit combined bars
4. Table 6 Combine Business and Management
5. Fig 10 - Not readable
6. Need some supporting cites and/or links to prior sections in the paper in the discussion section. I mostly agree w/ discussion but if one did not not it would be nice to have reference to foundation in the body of the paper.
Author Response
Dear reviewer.
Thank you so much for your kind comments, which are very useful for the improvement of our paper and research.
The paper has been exactly modified according to your advice. As for the last piece of advice, you are absolutely right that Fig.10 is not readable in the paper, since it has been compressed, but I think it also delivers some information, by being read in the original image.
Thank you again for your precious comments
Best,
Han
Reviewer 2 Report
The current research status and the historic developing trend of BI is very interesting and popular topic.
The research topic is related to the application of big data and predictive analytics to business intelligence. Research results are interesting and well presented. The most academic influential countries, institutions, journals, authors, and articles are identified.
It is well known that big data and artificial intelligence-based methods for BI will be one of the most popular research topics in the next few years, especially when it applies to industries related to topics of covid-19, healthcare, hospitality and 5G, but it is not clear how this paper can explore the future direction of business intelligence. Please, explain that.
Author Response
Dear reviewer,
Thank you so much for your kind comments, which are very useful for the improvement of our paper and research.
The conclusion of “It is well known that big data and artificial intelligence-based methods for BI will be one of the most popular research topics in the next few years, especially when it applies to industries related to topics of covid-19, healthcare, hospitality and 5G” is concluded by exploring the burst keywords and the corresponding references in 2021. Representative articles and conclusions are listed in Tab.11. The research could fall on such contents.
Covid-19: The major topics of Covid-19, big data, predictive analytics and BI research falls on the challenges and BI solution for firms due to the epidemic. The other topic is about the application of BI utilized in the research on influential effects of covid-19 on business industries.
Healthcare and hospital: The major topics is related to the challenges and advanced technologies of BI applied on healthcare industries. Cause the patient data safety and its application to business is one of the incentive topics, which should be discussed in the future.
Hospitality is one of the most influent industries by Covid-19, where BI and information technology-driven solution is the most effective and novel methods for levering the hospitality growth trend.
5G: 5G technology brings new opportunities for BI in the aspects of quality service monitoring, effective decision-making, efficient operation management, etc.
Thank you again for your precious comments
Best,
Han
Reviewer 3 Report
The submission attempts to pinpoint the current stance of AI in business analytics through a literature review scope. Although the topic is of great research interest, the manuscript is very badly written, poorly motivated and ill-presented. More specifically:
First and foremost, the manuscript shpuld be edited by a native English speaker, as it is full of expressional and grammar erorrs. Just a few example (not an exhaustive list):
The most academic contributed countries, institutions, cited authors, cited journals, and cited references are identified
With the increasingly generated data from internal and external platforms of the company, big data analysis becomes to be the most potential approach for BI insight extraction.
It aims to support managers to make reasonable decisions by predicting the future developing trend based on the historic collected databases
as the advanced technology of artificial intelligence (AI) exhibits significant advantages
Scopus is one of the word-leading citation databases, which ensures a high quality of publications by an independent strict content selection system
(burst in burst in 2012)
The use of the CiteSpace (5.3.R4, 64-bit) and JRE (1.8) is not motivated and not discribed. The reader should be keen with the specifics of the tools, the underlying methodology and the actual results that the specific tools produce to understand the significance of the results. Otherwise, everything is not comprehensible.
Following my previous point, the whole study seems to be based on arbitrary choices. Tables of results that are not motivated nor explained, their significance in left for the reader to find and their link to teh actual research field is simply unknown. A literature review should at least highlight the emerging trneds and how they change the field throgh time.
Under the same argument, the selection of time zones in the paper and of selected paper should be motivated.
My main concern lies with the selection of the literature pool. Using the entire pool in the analysis is far from warranted, since not all publications and journals have the same impact in the field. So, at least, a selection of the available studies based on the quality of the journals should be made. That is why universites and faculty with very low ranking in the global university list (pick any you like) are in t etop 5 and other departments, research institutes and tech firms are absent. It is stange for instance that Google, Microsoft, Moody's or other tech firms with very important tools in Business analytics and significant research are absent from the list.
Overall, the submission is badly written, lacks innovation, needs methodological inprovement and better motivation.
Author Response
Dear reviewer.
Thank you so much for your kind comments, which are very useful for the improvement of our paper and research.
The paper has been exactly modified according to your advice. And I am so sorry that the inconvenience caused by the typo. An introduction of the tools and more details of the explanation of tables and figures have been added or modified in this article.
You are right about the literature database, which has been added to the limitation of this article. The selected literation pool is set as Web of Science Core Collection (WoSCC) and Scopus. These two databases are involved as the major citation sources for biblio-graphic studies. I think other literation databases may be considered in the future. The most academically contributed countries, institutions, authors, disciplines, references, journals, and topics are all listed by the quantitative analysis results, like citation numbers. Results are data-driven.
As for your last precious comment, I guess that universities may be more focused on academic studies than Google, Microsoft, Moody's, and other tech firms. Otherwise, I totally agree that these companies make more contribution practically. That’s also a great direction for future research, which is also mentioned in the modified article.
Thank you again for your precious comments
Best,
Han
Reviewer 4 Report
The work is an interesting contribution on the bibliometrics of business intelligence. Its results, however, depend on a quite restrictive article search in WoS and Scopus, which leaves out important categories and reference textbooks. Second, the importance measures of papers are not mathematically defined and are not uniquely determined. Last, the paper contains many English mistakes and typos which should be corrected.
In more detail:
1. references. The authors use two set of keywords for the search: one for WoS and one for Scopus? why are they different ? eg scopes contains more keywords. More importantly: why they did not use "data mining" as a keyword? Data mining has been associated with business intelligence, particularly by two key textbooks, with many references in google scholar, WoS and Scopus, which should be cited and included in the paper, for completeness:
- David J. Hand, Heikki Mannila, Padhraic Smyth: Princliples of data mining, MIT press, 2001.
- Giudici, P. Applied data mining: statistical methods for business and industry, Wiley, 2003.
2- Importance measures. The authors use centrality measures, number of publications, betwenness to rank authors, countries, institutions and categories. They should mathematically define these measures and use them consistently. That is, choose one of them, such as centrality, and use it for all rankings. Centrality is a normalised measure, better than the number of papers. They should also use weighted centralities and not also degree centrality (similarly to the page rank algorithm)
3. Given the somehow arbitrary choice of keywords and measures, author ranks are bound to be unstable and should not be included
4. The authors should include journals that are field specific, and contain the keywords in their title. For example: Journal of Big data, Statistical analysis and data mining, Applied Stochastic models in business and industry, computational statistics and data analysis. If the authors do not include these field specific journals important reference papers may be omitted.
To summarise, the authors should substantially revise their paper and satisfactorily reply to the above comments, before the paper can be considered for publication
Author Response
Dear reviewer.
Thank you so much for your kind comments, which are very useful for the improvement of our paper and research.
The paper has been exactly modified according to your advice. You are right about the keywords, and it is mentioned in the literation that more keywords, especially for “data mining”, should be considered. And data mining is a tool for predictive analysis for BI, the two references have been mentioned in the article. As for “there are two sets of search strings for WOS and Scopus”, because of the different search patterns for these two databases, it is actually the same from the bottom but described in different ways.
You are right that the measurement is unstable. It has also been mentioned in the limitation that “the insights are extracted based on the results analyzed by the tool of CiteSpace, where papers are requested to be imported as the standard format, the information other than the imported citation is not able to be explored.” But I still think It may deliver reference value for future research related to big data and predictive analytics applied to BI.
Thank you again for your precious comments.
Although the Journal of Big data is not identified as the top 5 journals by the quantitative analysis results, but they are still on the cited journal list. Actually, 496 journals are included, the Journal of Big data is ranked at 76, cited by 21 searched publications.
Best,
Han
Round 2
Reviewer 3 Report
The submission still lacks all those elements that constitute candidancy for publication. The bibliometric tools are not presented, only mentioned as an R library, minor presentation interventions, while all my main comments remain unanswered. Thus, I cannot propose acceptance.
Author Response
Dear reviewer,
Thank you so much for your comments. They are very precious and important.
The tool of CiteSpace this study used for bibliographic analysis has been explained in the 2.2 analysis tools. Typical citations have been applied to support the software's effectiveness and authority.
"
2.2 Analysis tools
This study utilizes the tool of CiteSpace (5.3.R4, 64-bit)(C. Chen, 2006; W. Wang & Lu, 2020) and JRE (1.8)(Suliyanti & Sari, 2019) for the literature analysis, which could be download from the website of “https://sourceforge.net/projects/citespace/”. According to the “The CiteSpace Manual” released in 2014(C. Chen, 2014), CiteSpace I (C. Chen, 2004)and CiteSpace II (C. Chen, 2006) are the first and the second version of this tool. The initialpublication of CiteSpace I (C. Chen, 2004) and CiteSpace II(C. Chen, 2006) from Chen’s has been cited from Google Scholar for 1882 and 4347 times respectively (The retrieval time is 16th September 2022). CiteSpace is interactive software running based on JRE (1.8) environment, aiming to knowledge extraction, exploring academic achievements, in-depth knowledge graphic visualization, scientific review, and literature quantitative analysis(C. Chen, 2006, 2014; Synnestvedt, Chen, & Holmes, 2005; W. Wang & Lu, 2020). Developing trend of academic opinions based on time series, contributed scholars, institutions, journals, and countries, and discipline subjects are able to be identified and analyzed by using this software, which have widely been utilized in bibliographic studies(H. Wang, 2022a, 2022b; H. Wang & Li, 2022; W. Wang & Lu, 2020; Yao et al., 2020). CiteSpace generates social networks with nodes and links, which indicates the degree of cooperation between authors, institutions and countries(Yue Sun et al., 2021). The shape of the nodes reflects the influential degree of the author, citation, journal, institution, country, etc. The weight of lines represents the degree of betweenness among nodes. The centrality value reflects the significance degree of nodes, where nodes with a centrality ≥ 0.1 is regarded as the key nodes (Liu & Huang, 2008; Yue Sun et al., 2021).
"
And the writing has been checked again.
Thank you again for your valuable comments.
Best,
Han
Reviewer 4 Report
The authors took into account most of my comments. The paper still misses important references in the field of predictive analytics. In particular, the authors should complete the paper adding reference to the topic of explainable big data analytics and artificial intelligence, which is becoming a very key topic. Examples of relevant references to be included are:
Bracke, Philippe & Datta, Anupam & Jung, Carsten & Sen, Shayak, 2019. "Machine learning explainability in finance: an application to default risk analysis," Bank of England working papers 816, Bank of England.
Paolo Giudici, Emanuela Raffinetti. Shapley-Lorenz eXplainable Artificial Intelligence,Expert Systems with Applications,Volume 167,2021,
Author Response
Dear reviewer,
Thank you so much for your precious comments. The topic of risk issues of AI applied business and finance is a very interesting one. Explainable big data and AI approaches should be paid more attention, since without a high level of interpretability, transparency, and accuracy, the black-box AI prediction algorithms may cause a huge economic loss. The references and valuable opinions have been involved in the future direction recommendation part in the conclusion of this paper.
Again, thanks for your opinion so much.
Best,
Han